# Vestibular Rehabilitation Therapy and Corticosteroids for Vestibular Neuritis: A Systematic Review and Meta-Analysis of Randomized Controlled Trials

**DOI:** 10.3390/medicina58091221

**Published:** 2022-09-05

**Authors:** Hanik Badriyah Hidayati, Hana Aqilah Nur Imania, Dinda Sella Octaviana, Roy Bagus Kurniawan, Citrawati Dyah Kencono Wungu, Ni Nengah Rida Ariarini, Cempaka Thursina Srisetyaningrum, Delvac Oceandy

**Affiliations:** 1Department of Neurology, Faculty of Medicine, Universitas Airlangga, Surabaya 60132, Indonesia; 2Dr. Soetomo General Academic Hospital, Surabaya 60132, Indonesia; 3Faculty of Medicine, Universitas Airlangga, Surabaya 60132, Indonesia; 4Department of Physiology and Medical Biochemistry, Faculty of Medicine, Universitas Airlangga, Surabaya 60132, Indonesia; 5Department of Neurology, Faculty of Medicine, Universitas Indonesia, Jakarta 16424, Indonesia; 6Department of Neurology, Faculty of Medicine, Universitas Gadjah Mada, Yogyakarta 55281, Indonesia; 7Division of Cardiovascular Science, Manchester Academic Health Science Centre, University of Manchester, Manchester M13 9PG, UK

**Keywords:** vestibular neuritis, corticosteroid, vestibular rehabilitation, canal paresis, DHI

## Abstract

*Background and Objectives:* Besides corticosteroids, clinicians found that vestibular rehabilitation therapy (VRT) has a potential effect on vestibular neuritis (VN) improvement. This study aimed to investigate the efficacy of both corticosteroid therapy (CT) compared to VRT, and each group compared to their combination (CT vs. (CT+VRT) and VRT vs. (CT + VRT). *Materials and Methods*: Systematic searches were performed in PubMed, CINAHL, and Scopus for randomized controlled trials (RCTs) reporting the administration of at least CT and VRT for VN. The outcome of interest was VN’s subjective and objective improvement parameters. *Results:* Four RCTs involving a total of 182 patients with VN were eligible for systematic review and meta-analysis. The weighted mean difference (WMD) of canal paresis (objective parameter) in the CT group is significantly lower than in the VRT group after a 1 month follow-up (8.31; 95% CI: 0.29, −16.32; *p* = 0.04; fixed effect). Meanwhile, the WMD of Dizziness Handicap Inventory (DHI) (subjective parameter) in the VRT group is significantly lower than in the CT group after a 1 month follow-up (−3.95; 95% CI: −7.69, −0.21; *p* = 0.04; fixed effect). Similarly, the WMD of DHI in the combination group (CT+VRT) is significantly lower than in the CT group after a 3 month follow-up (3.15; 95% CI: 1.50, 4.80; *p* = 0.0002; fixed effect). However, there is no significant difference in all outcomes after 12 months of follow-ups in all groups (CT vs. VRT, CT vs. combination, and VRT vs. combination). *Conclusions*: This study indicates that CT enhances the earlier canal paresis improvement, as the objective parameter, while VRT gives the earlier DHI score improvement, as the subjective parameter. However, their long-term efficacy does not appear to be different. VRT has to be offered as the primary option for patients with VN, and corticosteroids can be added to provide better recovery in the absence of its contraindication. However, whether to choose VRT, CT, or its combination should be tailored to the patient’s condition. Future studies are still needed to revisit this issue, due to the small number of trials in this field. (PROSPERO ID: CRD42021220615).

## 1. Introduction

Vestibular neuritis (VN) is an acute disease characterized by prolonged spontaneous vertigo over a period of time. One of the three most common causes of peripheral vestibulopathy (the first is paroxysmal positional vertigo) is VN, with an incidence of 15.5 per 100,000 [1]. VN typically presents with acute dizziness, vertigo, nausea, vomiting, oscillopsia, and unsteadiness. In terms of numbers, vestibular neuritis accounts for 3.2–9% of patients attending clinics complaining of dizziness [2]. Data from outpatient clinics that specialize in the treatment of dizziness indicate that 7% of their patients have VN [3]. Since there are no confirmatory diagnostic tests, VN is a diagnosis of exclusion, based on the bedside and laboratory evaluation [4]. It is assumed to be of viral origin, in particular, because of the possible reactivation of latent herpes virus simplex type 1. However, recent studies show that the degree of latent infection of vestibular ganglia is less frequent than previously thought [5,6].

Even though the pathophysiology of VN remains unclear, Bronstein and Lempert add that asymmetric neuronal activity in the vestibular nucleus results in compensatory eye movement and posture adjustment so that the head feels like it is spinning [7]. If input from one side is stopped due to vestibular neuritis, the neuronal activity of the ipsilateral vestibular nucleus stops while the contralateral side is still active [7]. A retrospective study by Uffer et al. analyzed the pattern of vestibular lesions in 25 VN patients, and found that the pattern of lesions was varied and significant [8]. As many as 76% of cases mentioned did not lead to neuritis because there was no innervation pattern, another 24% had a definite pattern of lesions (16%), or perhaps (8%) supported the neuritis hypothesis.

There is no consensus guidance for the treatment of VN. Thus, the therapeutic options for VN vary from (1) corticosteroids, (2) antiviral therapy (acyclovir or valacyclovir hydrochloride), (3) a combination of corticosteroids, antiviral agents, and (4) vestibular exercise [9,10,11]. Treatment of VN consists of managing symptoms of vestibular neuritis, antiviral therapy (if a viral cause is suspected), and complementing a balance rehabilitation program. Corticosteroids are sometimes used as symptomatic therapy. However, the focus of steroid treatments is on symptom relief, as well as the use of antiemetic drugs to reduce nausea (ondansetron and metoclopramide). To reduce dizziness, drugs such as meclizine, diazepam, compazine, and lorazepam may also be prescribed by a doctor, while agents that suppress vestibular action should be used for no more than three days.

In the last two decades, vestibular rehabilitation therapy was introduced and is being studied as both adjunctive and substitutive therapy for VN. It exhibits equal efficacy to corticosteroids, and is even better at some points [12,13,14,15]. The justification for its clinical use in VN comes from its effect on promoting vestibular compensation mechanisms in active neuronal changes in the brainstem and cerebellum due to sensory conflicts from vestibular pathology [16]. A Cochrane review in 2015 also adds information on the availability of supporting moderate to solid evidence of its safety and efficacy in treating unilateral peripheral vestibular disorders, including VN [17]. The administration of vestibular rehabilitation therapy, therefore, could be considered by the clinician per each clinical setting encountered.

Meanwhile, the use of corticosteroids in vestibular neuritis remains controversial, despite their common use in VN. There is currently insufficient evidence for giving corticosteroids to patients with VN [4,11,12]. The use of corticosteroids in VN is usually used in the short term, and only in the acute phase. However, corticosteroids have adverse effects such as an immunosuppression effect, hypertension, hyperglycemia, dyspepsia, gastrointestinal bleeding, inhibition of wound repair, osteoporosis, metabolic disturbances, glaucoma, and cataracts [18,19]. Several studies on VN patients observed that VN patients who received corticosteroids reported hyperglycemia and diabetic destabilization as adverse effects of corticosteroids [12,18,20]. In addition, clinical recovery does not appear to be better in patients receiving corticosteroids, and, with so many adverse effects, corticosteroids can be contraindicated in certain patients [3]. Another approach to the management of VN, such as vestibular rehabilitation therapy, should be considered either as substitutive or additional therapy in patients with VN. A comprehensive review summarizing recent data is necessary to address this issue.

Therefore, the specific purpose of this systematic review and meta-analysis is to answer the following questions: Does vestibular rehabilitation therapy improve the clinical outcomes of patients with vestibular neuritis? Is vestibular rehabilitation therapy better than corticosteroid treatment in vestibular neuritis? Is the combination of vestibular rehabilitation therapy and corticosteroid treatment better than single types of those therapies?

## 2. Materials and Methods

This systematic review and meta-analysis were conducted per the guidance of the preferred reporting items for systematic review and meta-analysis (PRISMA) statement [21]. A detailed protocol was previously registered in PROSPERO (CRD42021220615).

### 2.1. Search Strategy

Systematic searches were performed by screening for qualified studies published up to 25 January 2021 through PubMed, Cumulative Index to Nursing and Allied Health Literature (CINAHL) via EBSCOhost, and Scopus databases, engaging keywords “vestibular neuritis”, “vestibular neuronitis”, “steroid”, “prednisolone”, “vestibular therapy”, “vestibular rehabilitation”, or their synonyms. The searching and screening process of appropriate studies was performed by three independent investigators (H.A.N.I., D.S.O., R.B.K.). Any discrepancies were discussed and resolved together among investigators (H.A.N.I., D.S.O., R.B.K.). The retrieved literature was limited to English-delivered studies in which any title or abstracts deemed potentially eligible for inclusion were obtained for full-text assessments.

### 2.2. Study Eligibility Criteria

Any study that satisfied the following criteria was included in this review: (1) randomized controlled trials (RCTs) evaluating administration of corticosteroids therapy (VRT), vestibular rehabilitation therapy (VRT), or both for vestibular neuritis (VN); (2) RCTs involving adults (>16 years old) without gender restriction, diagnosed with VN. Regarding types of intervention, this review included studies reporting administration of any corticosteroid (any timing, any dose, by oral/intravenous/intramuscular/intratympanic route, and of any duration), as well as medications such as prednisolone, dexamethasone, methylprednisolone, etc.; and vestibular rehabilitation therapy (any type, any timing, and of any duration). Meanwhile, any other cause of acute vertigo in the participant’s study (e.g., benign paroxysmal positional vertigo, Ménière’s disease) was excluded from this review.

### 2.3. Data Extraction and Quality Assessment

The following data were extracted from each included study: (1) first author’s last name and publication year; (2) patient characteristics: sample size, age range, and gender; (3) study characteristics: study design, intervention arms, and sample size allocation in each arm; (4) investigated outcomes. The outcome of interest in this study was improvement indicators of VN, both for the subjective parameters (e.g., Dizziness Handicap Inventory (DHI)) or objective evidence (e.g., canal paresis/caloric irrigation, otolith dysfunction recovery, vestibular-evoked myogenic potential (VEMP)). The included studies were further assessed for methodological quality using the risk of bias tool included in software RevMan version 5.4.1. (Cochrane Collaboration, London, UK). Data extraction and bias assessments were conducted by two independent authors (H.A.N.I. and D.S.O.), and any discrepancies were resolved by a third author (R.B.K.), also in an independent manner.

### 2.4. Data Statistics

The analysis was performed both descriptively and statistically whenever they complied with meta-analysis criteria. Meta-analysis was conducted to determine the pooled measure of subjective and objective parameters, by looking for the risk ratios or weighted mean difference from each eligible study. Either random effects or fixed effect models were used in accordance with the heterogeneity test result [22]. Heterogeneity was investigated with Cochran Q statistics (*p* < 0.10 indicated statistical heterogeneity) and I^2^ value, which is classified as negligible (0–25%), low (25–50%), moderate (50–75%), or high (>75%). In case of the presence of very high heterogeneity, we proceeded to perform a sub-groups meta-analysis, in order to explore possible factors causing this heterogeneity. Whenever adequate studies were retrieved, publication bias was examined both visually through Begg’s funnel plots [23], and statistically by employing Egger’s tests [23]. The meta-analysis was performed using RevMan version 5.4.1. (Cochrane Collaboration, London, UK) [24] to generate forest plots, pooled estimates, and the funnel plot. A *p*-value < 0.05 is deemed statistically significant.

## 3. Results

### 3.1. General Characteristics and Narrative Synthesis of Included Studies

The electronic search resulted in the initial identification of 5670 references. After the removal of duplicates, non-English studies, and an initial shift for relevance (wrong study type, non-open access study), we were left with 988 publications. We screened titles and abstracts of all 988 references, resulting in 28 potentially eligible articles. We obtained the full texts of all 28 studies and assessed them for eligibility. Four randomized controlled trials (RCT) met the inclusion criteria. In addition, we hand-searched the references of all the studies for which the full text was retrieved. However, we identified no additional studies that could provide data to answer the research question. A study flow diagram is shown in Figure 1, according to the template described in the PRISMA statement [21].

All included studies were assessed by the risk of bias tool included in software RevMan version 5.4.1, as presented in Figure 2. Full details are provided in Table 1 files 1. Overall, there are a total of 182 patients participating in the four studies, with a mean sample size of 37.25 and a range of 20 to 59. Patients were recruited from a variety of settings including emergency departments [12,13], the audiology unit of the hospital [14], and specialist centers in university hospitals [15]. The studies differed in the diagnostic criteria used to define the participants from vestibular neuritis patients.

#### 3.1.1. Participants

The eligible participants for this review were diagnosed with vestibular neuritis. A study by Goudakos et al. includes patients who presented with complaints of acute severe prolonged rotatory vertigo, nausea, vomiting, and postural instability with no recent auditory loss, no central lesion, ipsilateral horizontal semicircular canal deficit on head thrust test, horizontal spontaneous nystagmus with rotatory component away from affected ear, ot unilateral caloric lateralization ≥25% per Jongkees formula [12,25]. Those with glaucoma, chronic vestibular symptoms, severe hypotension, severe diabetes mellitus, and corticosteroid contraindication also satisfy the exclusion criteria. The baseline characteristics of involved participants are deemed statistically insignificant.

Similarly, Yoo et al., Ismail et al., and Tokle et al. exhibited quite similar criteria for eligible patients, except for the lower threshold for abnormal caloric lateralization in the studies by Yoo et al. and Ismail et al., which is ≥20% [12,13,14,15]. Baseline characteristics of compared groups in all studies are statistically significant.

#### 3.1.2. Intervention

Goudakos et al. aimed to compare the efficacy of vestibular rehabilitation to corticosteroid therapy for VN [12]. Investigated groups consisted of 3 weeks VRT compared to 25 day dexamethasone therapy (intravenous dexamethasone 24 mg/d for first 3 days, followed by 7 day tapering down, and 14 day daily regimen of oral dexamethasone of 2 mg/d after hospital discharge). Regardless of the studied arms, all patients receive dimenhydrinate 150 mg for a maximum of 3 days and a proton-pump inhibitor.

Yoo et al. were concerned with the efficacy of steroid (oral methylprednisolone) therapy among patients with VN receiving VRT [13]. In this study, all patients received VRT for at least a month until no VRT-evoked dizziness was observed. One interventional group received methylprednisolone 48 mg/d for 9 days, followed by a 5 day tapering off, and ranitidine bid for 14 day in this course. Of particular concern, both investigated groups in this study received Ginkgo biloba extract 80 mg bid (Ginexin-F^®^, SK Chemicals Life Science, Sungnam, Korea) for 4 weeks, and as-needed diazepam during the first 5 days of treatment.

In contrast to Yoo et al., Tokle et al. investigated the efficacy of VRT among VN patients receiving a ten day course of prednisolone [15]. VRT was continued until 12 months post-therapy, delivered by a combination of outpatient VRT (2 times/week for ten weeks; once weekly for 3–6 months; once monthly for 6–12 months) and home-based VRT (several times a day while experiencing dizziness and instability, with increased repetition gradually).

Furthermore, Ismail et al. observed the efficacy of corticosteroids, VRT, and their combination among patients with VN [14]. This study investigated three interventional arms at once, comprising of groups receiving six weeks VRT, two weeks oral methylprednisolone 20 mg tid with H2 blocker antiemetic, and another group with VRT and steroid therapy following each previous protocol.

#### 3.1.3. Outcomes

The presented outcomes were divided based on objective and subjective assessments. Four studies report DHI as a subjective measurement and three studies report canal paresis as an objective assessment. In the study reported by Goudakos et al., the administration of either oral dexamethasone or VRT in patients with acute VN shows equal improvements in patients’ subjective parameters, represented by the EEV and DHI score, even after follow-up for 12 months [12]. Similarly, canal paresis and otolith dysfunction recovery (VEMP) are insignificantly different between the corticosteroid and VRT groups. Regarding complete disease resolution, both the corticosteroid and VRT groups exhibit an equal disease resolution rate, even though patients receiving corticosteroids have a significantly higher disease resolution rate (*p* < 0.05) compared to those having VRT after a 6 month follow-up, suggesting corticosteroid’s enhancement on earlier disease recovery.

After having followed up at 1 month and 6 months, Yoo et al. observed that there are significant improvements in both subjective (DHI) and objective (caloric test, video head impulse test (vHIT), sensory organization test (SOT)) parameters compared to baseline characteristics [13]. Nevertheless, the addition of a steroid in the therapeutic strategy does not exhibit different improvement in contrast to the control arm that did not receive a steroid regimen.

Intriguingly, in a study by Tokle et al. with the setting of prednisolone as the standard of care (SoC) for VN, the addition of VRT in the remedial strategy results in a significant decrease in patients’ perceived dizziness score for movement-provoked dizziness, compared to those with receive prednisolone alone (15). However, there are no significant differences in self-reported symptom scores (DHI; Hospital Anxiety and Depression Scale (HADS); University of California Los Angeles Dizziness Questionnaire (UCLA-DQ); Visual Analog Scale-A (VAS-A), dizziness in certain movements/positions; Visual Analog Scale-B (VAS-B), dizziness at all times, even at rest; Visual Analog Scale-C (VAS-C], feeling of unsteadiness and imbalance while standing or walking; Vertigo Symptom Scale (VSS)) at the 3 month follow-up. Remarkably, this study reports that there are statistically significant differences in HADS (*p* = 0.039), DHI (*p* = 0.049), and VAS-C (*p* = 0.012) after a 12 month follow-up, favoring the VRT group.

Furthermore, a study by Ismail et al. involving three interventional arms of VRT, steroid therapy, and their combination, found equal improvements in both subjective (DHI score) and objective (caloric test and vestibular evoked myogenic potential (VEMP)) resolution indicators after having followed up at 1, 3, 6, and 12 months [14]. Nevertheless, regarding the improvement in the otolith dysfunction (VEMP), groups receiving steroid regimen exhibit better enhancement of otolith dysfunction resolution after 1, 3, and 6, month follow-ups compared to those having VRT only.

A comprehensive summary of extracted data of interest from the aforementioned RCTs is presented in Table 1.

### 3.2. Meta-Analysis of Vestibular Rehabilitation Therapy (VRT) vs. Corticosteroid Therapy (CT)

By involving a total of 40 patients, subjective (DHI score) and objective (canal paresis) parameters in patients with VN, either receiving VRT or CT strategy, could be quantitatively compared. A significantly lower DHI is observed in the VRT group (−3.95; 95% CI: −7.69, −0.21; *p* = 0.04; fixed effect) compared to the CT group at 1 month after follow-up, although the DHI score shows similar improvement after having followed up at 6 and 12 months. Furthermore, both VRT and CT groups exhibit an equal reduction in canal paresis at 6 and 12 month follow-ups, while the CT group shows significantly lower canal paresis compared to the VRT group at 1 month monitoring (8.31; 95% CI: 0.29, 16.32; *p* = 0.04; fixed effect). We also reported the equal otolith dysfunction (VEMP) recovery risk between both groups after 1 month and 6 month follow-ups. A full forest plot of the variable of interest is provided in Figure 3.

### 3.3. Meta-Analysis of Corticosteroid Therapy (CT) vs. Corticosteroid and Vestibular Rehabilitation Therapy Combination (CT + VRT)

A total of 53 samples from Tokle et al. and 40 samples from Ismail et al. were eligible to use in comparing DHI scores between CT and combination groups. Improvement in DHI score favors the patient group receiving combination therapy of VRT and CT, instead of corticosteroid only, after having followed up for 3 months (3.15; 95% CI: 1.50, 4.80; *p* = 0.0002; fixed effect) (Figure 4). However, a 12 month follow-up reveals an equivalent improvement in the parameter of interest (DHI). A full forest plot of the variable of interest is provided in Figure 4.

### 3.4. Meta-Analysis of Vestibular Rehabilitation Therapy (VRT) vs. Corticosteroid and Vestibular Rehabilitation Therapy Combination (CT + VRT)

Regarding this interest, two studies with 69 pooled patients meet the criteria for quantitative analysis. However, the pooled data of canal paresis and DHI scores of the investigated groups does not differ significantly, even after follow-up at 1 and 6 months after initial therapy (*p* > 0.05). A full forest plot of the variables of interest is provided in Figure 5.

## 4. Discussion

The current systematic review and meta-analysis report that corticosteroids seem to be comparable to VRT. In addition, patients undergoing VRT show a better improvement in the subjective parameter evaluated by DHI, whereas the objective parameter of canal paresis apparently becomes better in patients taking corticosteroids. Both results are observed a month after admission.

In some recent studies, VRT has become one of several options for VN management, as it aims to reduce dizziness and improve balance, as well as overall physical function [26,27]. VRT has also shown good results in the treatment of VN in many studies [28,29,30]. It was found that VRT succeeded in decreasing motion sensitivity (habituation exercise), improving gaze stability (eye–head movement exercises), balance, and increasing endurance in patients with VN [28]. Specifically, Tokle et al. state that VRT shows a significant improvement in some parameters, including DHI score, compared to the standard care group receiving corticosteroids [15]. VRT also appears to be one of safe approaches for treating VN. Of all included studies in this review, three studies report the adverse event outcome, and found no adverse event observed following VRT administration [12,13,15]. Instead, Goudakos et al. found a steroid-receiving patient with controlled diabetes mellitus developed disease destabilization and hyperglycemia, though it could be addressed by adjusting the steroid dose [12]. Yoo et al. also observed minor transient side effects such as dyspepsia, minor facial swelling, and mood swings [13].

This review observed a comparison of the results between three parameters: canal paresis, otolith dysfunction (measured by VEMP) recovery, and DHI. Seemingly, subjective parameters, such as DHI, might be an important consideration for improvement in parameters, as they represent the functional status of patients that possibly have activity limitations and participation restrictions due to VN, compared to vestibular tests (e.g., canal paresis, VEMP).

According to the previous study by Fishman et al., the initial goal of this systematic review and meta-analysis was also to provide an update on corticosteroid administration in patients with VN [31]. However, there is a lack of new trials observing the efficacy of corticosteroids to improve patients’ clinical symptoms in the last 10 years. It changed the study objective to the efficacy of corticosteroid therapy compared to vestibular rehabilitation therapy and each group compared to the combination of both therapies, since growing evidence suggests a promising result for VN management. Eventually, the renewal objective opened a potential new outcome from those four included studies.

The included studies by Goudakos et al. and Ismail et al. show a lower canal paresis within the first month after initiation of CT compared to VRT [12,14]. However, there is no significant difference in canal paresis improvement between CT and VRT at 6 and 12 month follow-ups. On the other hand, the DHI score in the VRT group is significantly lower at a month of monitoring than in the CT group, while DHI at 6 and 12 months is balanced between the two groups. Yoo et al., separately, adds that both corticosteroid plus VRT and VRT-only groups show an equal disease resolution rate, even though patients receiving corticosteroids have a significantly higher disease resolution rate (*p* < 0.05) compared to those having VRT at a 6 month follow-up [13]. Similarly, Ismail et al. observed that groups receiving CT or combination therapy (CT+VRT) exhibit better enhancement of otolith dysfunction resolution at 1, 3, and 6 month follow-ups compared to those only having VRT, while the pooled risk ratio for otolith dysfunction (VEMP) recovery is equal in both groups [12,14]. The quantitative assessment captured the earlier resolution of the canal paresis in the CT group and the DHI score in the VRT group, though both groups exhibit equivalent outcomes at a longer time point.

Canal paresis observed by the caloric test is one of the diagnostic criteria for VN, and its improvement is extensively analyzed as an objective parameter of VN recovery [32]. Canal paresis is a condition in which the inner ear’s labyrinthine system fails to respond to caloric test stimulus on the affected side [33]. It is an important finding in dizzy patients, including in the setting of vestibular neuritis [33]. Complimentary to canal paresis, VEMP was introduced as a vestibular function test, particularly assessing saccular function [34,35]. In peripheral vestibular dysfunction, an absence/reduction in VEMP amplitude is expected. VN with superior branch involvement, canal paresis, and normal VEMP might be observed, while VN with inferior branch involvement results in a normal caloric test and abnormal VEMP. In our meta-analysis, we documented earlier canal paresis improvement in the CT compared to the VRT group. This might be partly explained by the effect of steroids to reduce nerve inflammation, whose efficacy is documented in the setting of various acute peripheral neuritis, providing a rationale for its clinical use in VN management [12]. An insignificant difference in the pooled risk ratio of otolith dysfunction recovery, however, still needs to be further explored. That the number of patients who developed abnormal VEMP is too small provides a significant limitation to make a conclusion, and may introduce a bias during interpretation. Both Goudakos et al. and Ismail et al. only measure otolith dysfunction by the cervical VEMP (cVEMP) test, which specifically assesses the tract from the sacculus via the inferior branch of the vestibular nerve [12,14]. The involvement of this branch is considerably less observed compared to the superior branch. Meanwhile, superior branch involvement might also present with otolith dysfunction, which is documented by the ocular VEMP (oVEMP) test. To date, studies reporting the evaluation of CT and/or VRT for the recovery of oVEMP in patients with vestibular neuritis are lacking. Therefore, future studies specifically concerning vestibular nerve branch involvement and each response (subjectively and objectively) to CT and/or VRT are encouraged to be conducted, in order to provide evidence of whether specific nerve branch involvement affects the outcome of therapy.

Meanwhile, DHI, which stands for Dizziness Handicap Inventory, is one of the most popular questionnaires for the assessment of dizziness [36]. DHI seems to be more reflective of the patient’s real condition, as these questionnaires might be more sensitive than the others for assessing long-term effects on the emotional and psychological well-being of patients with VN [15].

Moreover, this review tried comparing the DHI score of patients receiving corticosteroids only with a combination of VRT and CT. The DHI score in the combination group exhibits better improvement at 3 months of follow-up therapy. Moreover, Tokle et al., individually, report that there are statistical differences in HADS, DHI, and VAS-C after a 12 month follow-up, favoring the VRT-receiving group [15]. This lower DHI score might be due to the VRT presence, since the patients might experience lower stress levels compared to patients who only take medication (corticosteroid). Previous clinical research found that, compared with drug therapy alone, patients receiving vestibular rehabilitation and also drug therapy show significant improvement either in vertigo and disability symptoms, or in anxiety and depression after 1 month of rehabilitation [37,38,39].

Prior studies found that patients with acute vestibular dysfunction experience more anxiety and depression, which can cause the sensation of dizziness [40,41,42,43,44,45,46,47,48]. A study by Teggy et al., comparing DHI scores of VN patients administered with VRT and those without VRT, specifically found that the emotional score in DHI is lower than the physical and functional score [28]. This result is probably related to the psychological aspect of patients, which is associated with the occurrence of dizziness and the patient’s quality of life [46,47,48,49]. In conclusion, emotional and psychological well-being is found to play a major role in persisting dizziness after vestibular neuritis, which is an important thing related to patients’ normal activities.

The present review also endeavored to compare the administration of VRT alone with the combination of VRT with CT. The result indicates that both canal paresis and DHI score in both arms do not show significant outcomes. Thus, adding a corticosteroid regimen in VN patients treated with VRT might not enhance disease recovery. The study by Goudakos et al. supports the idea that patients treated with corticosteroids have no advantage in the long-term prognosis of their disease, since it only has an anti-inflammatory and anti-edema effect, which is less specific for the source of the VN disease [12,16]. The result seems to be confirmed by the previous meta-analysis comparing corticosteroids with placeboes, which also shows evidence of insignificant differences in clinical symptoms recovery and DHI score after a month [3]. Indeed, VRT has a role in accelerating central compensation through a habitual training mechanism, as well as increasing substitution. It provides four targets in the form of increased stability, vision, postural stability, vertigo symptoms, and improvement in daily life activities [16,50].

In the end, it should be noted that this study’s results possess several limitations. Even though this review includes the most recent RCT by Tokle et al. in 2020, the number of eligible RCTs is still limited, resulting in fewer pooled data. Several included RCTs possess a high risk of bias concerns, suggesting clinicians carefully interpret results. These few studies also limited the opportunity to compare all parameters at each follow-up time. The influence of the diversities of duration and type of VRT delivery, and type and doses of corticosteroids used on the pooled outcome have not been assessed, due to an inadequate study number. Yet, there is not one specific type of VRT for an optimal recovery because this intervention is given with consideration of various factors such as pathology, age, motivation, and reduction, all within the environmental context [51]. In the future, choosing the VRT is bound by the patient’s physical condition, such as sensorimotor, cognitive, emotional, and psychosocial profiles. It indicates that more well-designed RCT concerning this aspect must be carried out in the future. However, this present review might add novel evidence in the field of VN. To the best of our knowledge, this review employed the most recent literature search time periods, which included RCT by Tokle et al. published in 2020, allowing us to perform further comparative analyses of CT vs. combination and VRT vs. combination group.

## 5. Conclusions

The present review reports that the VRT seems to be as good as the CT. VRT promotes earlier DHI improvement, while CT enhances earlier canal paresis improvement. In the setting of CT as standard therapy, the addition of VRT significantly improves DHI as the subjective outcome. Nonetheless, the long-term efficacy of CT and VRT do not appear to be different. Future well-designed trials are still needed, and a clear interim guideline should be arranged. Our study might add that VRT has to be offered as the primary option for patients with VN, and corticosteroids can be added to provide better recovery in the absence of its contraindication. However, whether to choose VRT, CT, or its combination, should be tailored to the patient’s condition and health status.

## Figures and Tables

**Figure 1 medicina-58-01221-f001:**
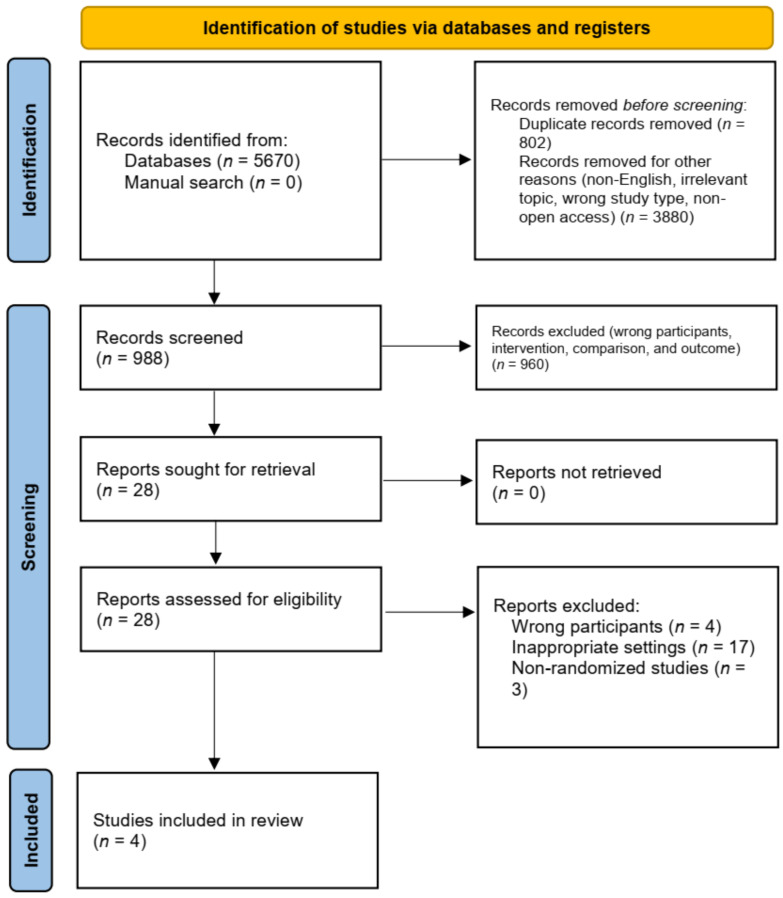
PRISMA flow diagram illustrating the literature search process.

**Figure 2 medicina-58-01221-f002:**
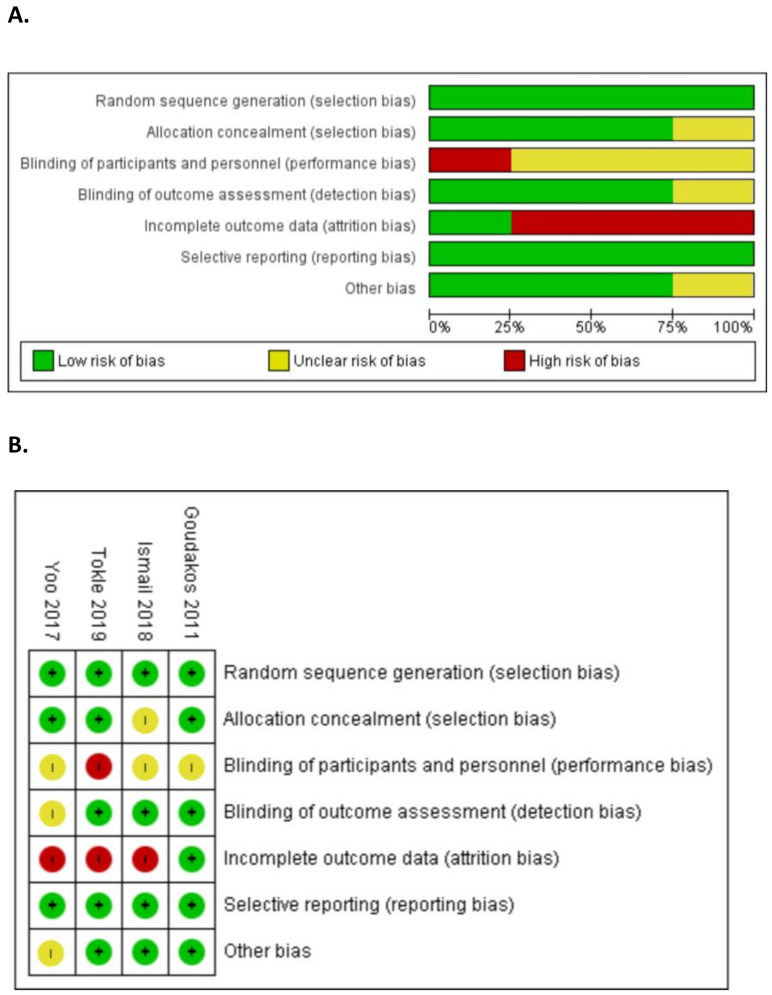
Risk of bias analysis results. (**A**) Risk of bias summary: review authors’ judgments about each risk of bias item for each included study. (**B**) Risk of bias graph: review authors’ judgments about each risk of bias item presented as percentages across all included studies [12,13,14,15].

**Figure 3 medicina-58-01221-f003:**
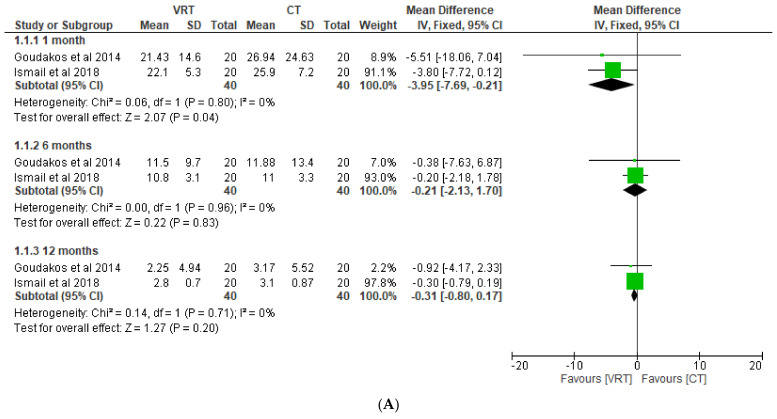
Forest plots of DHI (**A**) and canal paresis (**B**) comparison between VRT and CT groups at 1 month, 6 months, and 12 months; otolith dysfunction recovery risk ratio (**C**) between VRT and CT groups at 1 month and 6 months [12,14].

**Figure 4 medicina-58-01221-f004:**
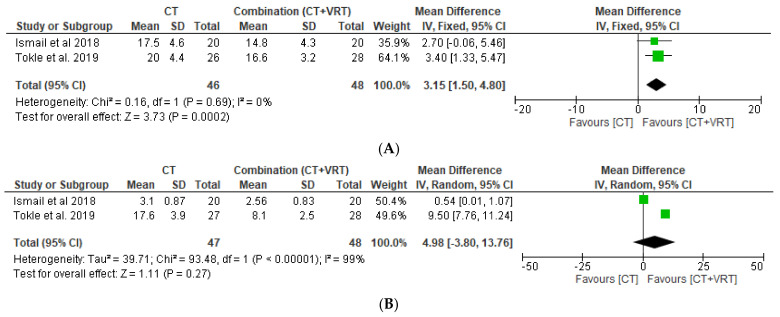
Forest plots of comparison of DHI score at (**A**) 3 months, and (**B**) 12 months in CT and combination groups [14,15].

**Figure 5 medicina-58-01221-f005:**
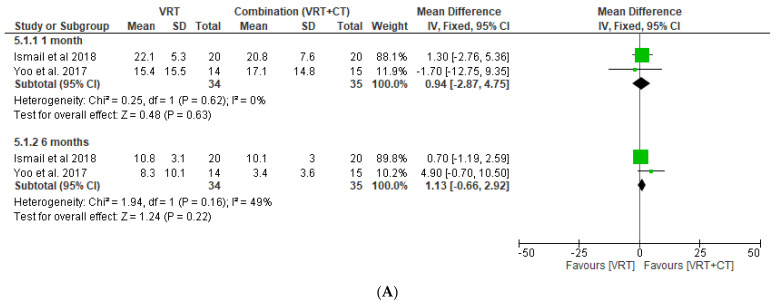
Forest plots of comparison of DHI score at 1 month and 6 months (**A**); and canal paresis at 1 month and 6 months (**B**) in VRT and combination groups [13,14].

**Table 1 medicina-58-01221-t001:** Characteristics and Findings of Involved Studies.

Author, Publication Year	Study Design	Population	Sample Size			Standard of Care (SoC)	Studied Arms			Primary Outcomes	Follow Up, Months	Findings
			CT	VRT	Combination		CT	VRT	Combination			
Goudakos et al., 2014 [12]	RCT	Patients (18–80 years old) with VN	20	20	NR	Dimenhydrinate Proton pump inhibitor SoC for 3 days	25 days dexamethasone	3 weeks VRT	NR	Clinical (EEV, DHI), canal paresis, otolith (VEMP) recovery	1, 6, 12	Both arms exhibit equal improvement in clinical, canal, and otolith parameters, ns Better canal paresis recovery in the first month (CT vs. VR, 0/20 vs. 2/20) Significantly higher ratio of CT group with complete disease resolution at 6 month follow-up (*p* < 0.05)
Yoo et al., 2016 ^a^ [13]	RCT	Patients (19–80 years old) with VN	NR	15	14	*Ginkgo biloba* extract, intravenous or oral diazepam VRT for at least a month until no VRT-evoked dizziness is observed.	NR	As SoC	14 days methylprednisolone 14 days ranitidine VRT as SoC	Improvement in objective (CT, SOT, vHIT) and subjective (DHI) parameters	1, 6	Equal improvement in objective parameters in both groups, ns Equal improvement in subjective parameters in both groups, ns
Ismail et al., 2018 [14]	RCT	Patients (20–50 years old) with VN	20	20	20	Dimenhydrinate for a maximum of 3 days	2 weeks methylprednisolone, H2 blocker	6 weeks VRT	CT + VRT protocol	Improvement in objective (CT, VEMP) and subjective (DHI) indicators	1, 3, 6, 12	Equal improvement in objective parameters in both groups, ns Equal improvement in subjective parameters in both groups, ns
Tokle et al., 2020 ^a^ [15]	RCT	Patients (18–70 years) with VN	27	NR	3 months 27 12 months 26	10 days prednisolone	As SoC	NR	12 months VRT + SoC	Perceived dizziness during head motion, HADS, DHI, UCLA-DQ, VAS, walking speed, and standing balance	3, 12	Significant improvement in overall perceived dizziness score, favoring toward combination group at 3 and 12 month follow-ups (*p* = 0.007, *p* = 0.001) Equal improvement in HADS, DHI, UCLA-DQ, VAS, walking speed, and standing balance, ns

NR, not relevant; ns, not significant (*p* > 0,05); ^a^ loss to follow-up issue, Yoo et al.: 3 patients in each group (formerly 18 patients for combination group and 17 for VRT group), Tokle et al.: 1 patient in combination group at a year and 2 patients in CT group at 3 months (formerly 27 patients for combination group and 29 patients for CT group).

## Data Availability

Data are contained in the article, and additional data are available upon request from the corresponding author.

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
