# Peer review of "Vestibular Rehabilitation Therapy and Corticosteroids for Vestibular Neuritis: A Systematic Review and Meta-Analysis of Randomized Controlled Trials"

_medicina, 2022, doi:10.3390/medicina58091221_

Round 1

Reviewer 1 Report

Abstract: In the abstract, you have to write VN line 19 as Vestibular Neuritis. 

Introduction: The introduction is clear. However, you introduce the side effects of corticosteroids but do not debate them in the discussion extensively. It would be good if you could also show adverse reactions to corticosteroids, in the study you mentioned if they are present. Despite the improvement of overall DHI. 

Results: I would consider also gender differences and compare gender and their recovery period if you have these data. If you have any data about comorbidities I would present them. I would summarize the results as much as possible in fewer figures. Comparing clearly VRT and corticosteroids study in 1 graph showing the DHI score and follow up. I would also show in a graph the differences between canal paresis and otolith dysfunction recovery. 

Discussion: I would debate if the studies using corticosteroids are considering side effects if there are age or gender response differences. Also if the VRT studies are reporting long-term follow-up. Do you know if any of these patients further developed Visually Induced DIzziness or Visual Motion Sensitivity? Can you also comment on the differences on canal paresis and otolith dysfunction recovery?

Conclusion: I would make the remark about the need for clear guidelines in the management of VN and what would be the suggestion from the analysis you performed. If a combination of corticosteroids and VRT seems the most appropriate and why? 

Author Response

Dear Editor and Reviewers,
We would like to thank the reviewers for careful reviews and constructive suggestions to
our manuscript. We have revised the manuscript in keeping with reviewers’ comments and
suggestions and detailed replies are hereby attached.

We hope the manuscript will be suitable for Medicina readers’ interest and look forward to
receiving from you

Reviewer 2 Report

In your introduction, you should describe shortly what is the rational of VRT.

The section (line 72-89) on corticosteroid is too long. As an example, growth retardation of children (line 78) . VN is rarely found in children. Also, i would never give steroids in VN in a diabetic patient.  This chapter can be much shorter .

In this way, a section with VRT and a second section with CT, the objectives of your study would be much better highlighted. 

Line 354 "zin" should be "within"

In your conclusion, i would insist on VRT and your last sentence is quite elusive (line 430) . You should take position based on this meta analysis and conclude that the modern treatment should be mainly VRT.

Author Response

Dear Editor and Reviewers,
We would like to thank the reviewers for careful reviews and constructive suggestions to
our manuscript. We have revised the manuscript in keeping with reviewers’ comments and
suggestions and detailed replies are hereby attached.

We hope the manuscript will be suitable for Medicina readers’ interest and look forward to
receiving from you.

Round 2

Reviewer 1 Report

Nice much better overall.

One question why is that in line 443?However, an insignificant difference in the pooled 439 risk ratio of otolith dysfunction recovery still needs to be explored, particularly whether 440 any selectivity of vestibular nerve branch involvement during VN affected steroid 441 responses. Indeed, only a small portion of VN patients in this review developed abnormal ....

I would expand on this part what do you think should be a next step?

Author Response

Dear reviewer, 

We would like to thank you for your constructive suggestion. Regarding your suggestion: 

Nice much better overall.

One question why is that in line 443? However, an insignificant difference in the pooled risk ratio of otolith dysfunction recovery still needs to be explored, particularly whether any selectivity of vestibular nerve branch involvement during VN affected steroid responses. Indeed, only a small portion of VN patients in this review developed abnormal...

I would expand on this part what do you think should be the next step?

The authors of two studies included in the VEMP meta-analysis (Goudakos et al. and Ismail et al.) also did not explain their VEMP finding in their discussion. This is possibly caused by the small number of subjects with abnormal VEMP which limit the authors of those studies to conclude the finding. We have addressed your suggestion by providing a revision in line 439,

An insignificant difference in the pooled risk ratio of otolith dysfunction recovery, however, still needs to be furtherly explored. That the number of patients who developed abnormal VEMP is too small provides a significant limitation to take a conclusion and may introduce a bias during interpretation. Both Goudakos et al. and Ismail et al. only measured otolith dysfunction by cervical VEMP (cVEMP) test which specifically assesses the tract from the sacculus via the inferior branch of the vestibular nerve. Involvement of this branch is considerably less observed compared to the superior branch. Meanwhile, superior branch involvement might also present with otolith dysfunction which is documented by the ocular VEMP (oVEMP) test. To date, studies reporting the evaluation of CT and/or VRT for the recovery of oVEMP in patients with vestibular neuritis are lacking. Therefore, future studies specifically concerning vestibular nerve branch involvement and each response (subjectively and objectively) to CT and/or VRT are encouraged to be conducted to provide evidence of whether specific nerve branch involvement affects the outcome of therapy.

We hope the manuscript will be more improved and suitable for Medicina readers’ interest. We look forward to receiving from you.

Your sincerely,

Hanik Badriyah Hidayati, M.D., Ph.D.

Department of Neurology, Faculty of Medicine, Universitas Airlangga, Surabaya, Indonesia/Dr. Soetomo General Academic Hospital, Surabaya, Indonesia.

On behalf of all authors.